# Fairness and robustness in anti-causal prediction

**Maggie Makar**                                                 *mmakar@umich.edu*
*Computer Science and Engineering*
*University of Michigan*
*Ann Arbor, MI*

**Alexander D'Amour**                                            *alexdamour@google.com*
*Google Research*
*Cambridge, MA*

**Reviewed on OpenReview:** *https://openreview.net/forum?id=EhOUSGlIxXt*

## Abstract

Robustness to distribution shift and fairness have independently emerged as two important desiderata required of modern machine learning models. While these two desiderata seem related, the connection between them is often unclear in practice. Here, we discuss these connections through a causal lens, focusing on anti-causal prediction tasks, where the input to a classifier (e.g., an image) is assumed to be generated as a function of the target label and the protected attribute. By taking this perspective, we draw explicit connections between a common fairness criterion—separation—and a common notion of robustness—risk invariance. These connections provide new motivation for applying the separation criterion in anticausal settings, and inform old discussions regarding fairness-performance tradeoffs. In addition, our findings suggest that robustness-motivated approaches can be used to enforce separation, and that they often work better in practice than methods designed to directly enforce separation. Using a medical dataset, we empirically validate our findings on the task of detecting pneumonia from X-rays, in a setting where differences in prevalence across sex groups motivates a fairness mitigation. Our findings highlight the importance of considering causal structure when choosing and enforcing fairness criteria.

## 1 Introduction

In real world applications of machine learning, there is a strong desire to enforce the intuitive criterion that models should not depend inappropriately on sensitive attributes. Such a desire is often formalized in terms of specific quantitative notions of fairness. Fairness criteria often focus on equalizing model behavior across different population subgroups defined by a sensitive attribute. When designing fair systems, practitioners face a number of choices: among them are the choice of fairness criterion, and how to implement it. For example, a practitioner may wish to choose between simply using an unconstrained model that maximizes overall performance, a model that makes predictions independently of the sensitive groups (independence), or a model that makes predictions independently of sensitive groups given the ground truth label (separation) (see Barocas et al., 2019, Chapter 2).

Here, we show that the causal structure of a problem can be a useful piece of context for choosing and enforcing a fairness criterion in a given application. We present this argument by making a connection to some recent insights on the relevance of causal structure to robust machine learning. In principle the connection between robustness and fairness should not be surprising. Similar to fairness methods, robustness methods also seek to regulate how classifiers depend on particular features; however, the details of the connection depend on the particular causal structure of the problem at hand. In particular, we focus on an anti-causal prediction setting, where the input to a classifier (e.g., an image) is assumed to be generated as a function

of the label and the sensitive attribute (see Figure 1). In this context, we show that the connection between fairness and robustness can provide new motivations and methods for enforcing fairness criteria in practice.

Concretely, we use the example of detecting pneumonia from chest X-rays as a motivating example, inspired by Jabbour et al. (2020). In this example, we assume that both the presence of pneumonia ($Y$) and the patient's sex ($V$) causally influence the X-ray image ($\mathbf{X}$) used to make the diagnosis. In this context, practitioners may aim to make diagnoses that respect certain fairness criteria across sex groups. As we show, under the causal structure of this problem, some of these fairness criteria map neatly onto robustness criteria that seek to preclude the classifier from using image features influenced only by $V$.

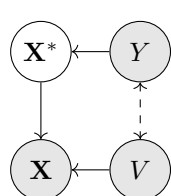

Figure 1: Causal DAG of the setting in this paper. The main label $Y$ and the sensitive attribute $V$ generate observed input $\mathbf{X}$, but $Y$ only affects $\mathbf{X}$ through the sufficient statistic $\mathbf{X}^*$.

Informed by this causal structure, we provide a description of how fairness criteria and distributional robustness criteria align, and discuss practical implications of this alignment for motivating and enforcing fairness criteria in this setting. Specifically, we focus on alignment between the separation fairness criterion—that the distributions of predictions within the positive and negative classes should be independent of the sensitive group—and risk invariance as a robustness criterion—that the predictive risk of a model remain invariant across a family of distribution shifts. We show that:

- The separation fairness criterion implies risk invariance across a family of distribution shifts that change the base rate of the label $Y$ within sensitive groups $V$, but respect the causal structure of the problem. This provides new perspective on discussions of fairness-performance tradeoffs when applying the separation criterion in practice.
- In practice, algorithms designed to enforce risk invariance also enforce the separation criterion, in some cases more effectively than algorithms that attempt to directly incorporate the separation criterion as a regularizer. We explore properties of one particular algorithm, presented in Makar et al. (2021), which uses a weighted regularizer based on the maximum mean discrepancy (MMD), and compare it to an MMD-based implementation of an empirical separation criterion.
- For contrast, we show a conflict between the independence fairness criterion (often measured in terms of equalized predictions between $V$ groups, or demographic parity) and the risk invariance property.

A number of the connections that we describe here between fairness, robustness, and causal structure have been noted before in work aimed at improving robustness under input perturbations or distribution shift, most notably Veitch et al. (2021) and Makar et al. (2021). Our main contribution is to draw out the practical implications of these results for fairness in a concrete setting. Our findings support paying increased attention to the causal structure of a problem to inform the choice and implementation of fairness criteria in practice, even if these metrics are on their face "oblivious", or independent of causal structure. This causal focus can have the advantage of highlighting connections to other desiderata and methodology.

## 2 Related work

Robustness to distribution shift and fairness are closely related, and many lines of work have aimed to highlight formal and empirical connections between them. For example, Sagawa et al. (2019) explored applying distributionally robust optimization to address worst-subgroup performance for under-represented groups. As another example, Adragna et al. (2020) show that using methods meant to induce robustness leads to "more fair" classifiers for internet comment toxicity. Along similar lines, Pruksachatkun et al. (2021) found that certified robustness approaches designed to ensure robustness of NLP methods against word substitution attacks can be used to reduce violations to the equalized odds criterion.

A key perspective in our work is that many fairness-robustness relationships are mediated by causal structure. In this sense, our work is most similar and complementary to Veitch et al. (2021), in which the authors derive implications of counterfactual invariance to certain input perturbations, and show that these implications depend strongly on the causal structure of the problem at hand. Here, we focus on a narrower setting,

and highlight concrete conclusions about the relationship between easily measurable robustness and fairness metrics that are often used for evaluation in practice, with a greater focus on implications for fairness.

Notably, our use of causal ideas is distinct from another body of work that defines fairness criteria directly in terms of the causal model. These include definitions of fairness that revolve around direct causal effects of sensitive attributes on outcomes (Kilbertus et al., 2017; Nabi & Shpitser, 2018; Zhang & Bareinboim, 2018), or discrepancies between counterfactual outcomes (Kusner et al., 2017; Chiappa, 2019). By contrast, we focus on "oblivious" fairness criteria that are not themselves a function of causal structure (Hardt et al., 2016), but show that the causal structure of a problem can still inform when and how to use such a criterion.

## 3 Background and preliminaries

**Setup**  We consider a supervised learning setup where the task is to construct a predictor $f(\mathbf{X})$ that predicts a label $Y$ (e.g., pneumonia) from an input $\mathbf{X}$ (e.g., chest X-ray). In addition, we have a protected attribute $V$ (e.g., patient sex) available only at training time. Throughout, we will use capital letters to denote variables, and small letters to denote their value. Our training data consist of tuples $\mathcal{D} = \{(\mathbf{x}_i, y_i, v_i)\}_{i=1}^n$ drawn from a source training distribution $P_s$. We restrict our focus to the case where $Y$ and $V$ are binary and $f$ is a classifier. Specifically, we will consider functions $f$ of the form $f = h(\phi(\mathbf{x}))$, where $\phi$ is a representation mapping and $h$ is the final classifier.

In this context, a practitioner may be interested in ensuring that the classifier $f(\mathbf{X})$ treats individuals from different groups fairly. Fairness is operationalized by enforcing constraints on how the distribution of model predictions can differ across individuals from different groups (that is, with different values of $V$) or individuals in different label classes (that is, with different values of $Y$).

We focus on two well-established fairness criteria (see Barocas et al., 2019, Chapter 2 for a full discussion). While these criteria are typically defined with respect to the predicted class (i.e., $\hat{Y} = \mathbb{1}\{f(\mathbf{X}) > \delta\}$ for some threshold $\delta$) we consider stronger fairness notions defined with respect to the predicted probabilities $f(\mathbf{X})$ (see "Analogues with Scores" in Mitchell et al., 2021). This focuses the exposition on issues relating to the quality of $f(\mathbf{X})$ independently of the choice of $\delta$.

The first criterion, separation, requires that the distribution of predictions $f(\mathbf{X})$ be the same across groups $V$ conditional on the ground truth label $Y$, that is $f(\mathbf{X}) \perp\!\!\!\perp V \mid Y$. Separation is often evaluated in terms of equalized odds (EO) (Hardt et al., 2016), which examines whether the conditional expectations of the predictions for each ground truth class $Y$ are the same across sensitive groups $V$, i.e., that

$$\mathbb{E}_{P_s}[f(\mathbf{X}) \mid V = 0, Y = y] = \mathbb{E}_{P_s}[f(\mathbf{X}) \mid V = 1, Y = y] \quad \forall y \in \{0, 1\}.$$

The second criterion, independence, requires that the distribution of predictions be the same overall across groups $V$. Independence is often evaluated in terms of demographic parity (DP), which examines whether the expected prediction is the same across groups, i.e., that

$$\mathbb{E}_{P_s}[f(\mathbf{X}) \mid V = 0] = \mathbb{E}_{P_s}[f(\mathbf{X}) \mid V = 1].$$

Unlike EO which requires equality conditional on $Y$, DP requires a marginal form of equality.

In practice, separation and independence are often enforced at training time using empirical measures of distributional discrepancies, while the effectiveness of these interventions are often evaluated by measuring how closely the model respects the EO and DP criteria on test data.

**Causal assumptions**  The separation and independence criteria are "oblivious" criteria: they are constraints on the observable distributions of labels and predictions that do not depend on the form of the classifier or the mechanisms of the data generating process (Hardt et al., 2016). However, the implications of enforcing these criteria do depend on the causal structure of the problem. Here, we outline the causal assumptions that underlie the anti-causal prediction setting that we study, and show how it connects to our running example of detecting pneumonia from chest X-rays.

We assume that $P_s$ has a generative structure shown in Figure 1, in which the inputs $\mathbf{X}$ are generated by the labels $(Y, V)$. We assume that the labels $Y$ and $V$ are are correlated, but not causally related; that is, an intervention on $V$ does not imply a change in the distribution of $Y$, and vice versa. Such correlation often arises through the influence of an unobserved third variable such as the environment from which the data is collected. We represent this in Figure 1 with the dashed bidirectional arrow.

In addition to the specific DAG, we assume that there is a sufficient statistic $\mathbf{X}^*$ such that $Y$ only affects $\mathbf{X}$ through $\mathbf{X}^*$, and $\mathbf{X}^*$ can be fully recovered from $\mathbf{X}$ via the function $\mathbf{X}^* := e(\mathbf{X})$. However, we assume that the sufficient reduction $e(\mathbf{X})$ is unknown, so we denote $\mathbf{X}^*$ as unobserved in Figure 1.

While anti-causal and sufficiency assumptions are somewhat strong, they are also reasonable in a number of important contexts where machine learning is applied. For example, the chest X-ray example plausibly satisfies these conditions. First, the disease (pneumonia) is the true cause of abnormal findings reflected in the chest X-ray. Secondly, there are few interactions between the presentation of pneumonia and sex in the radiograph, so it is plausible to assume that the sufficient features of the X-ray that are influenced by pneumonia $\mathbf{X}^*$ can be recovered from the input image $\mathbf{X}$. More generally, these assumptions are likely to apply in settings where the goal is to recover a pre-existing ground truth label $Y$ from an input $X$, and $Y$ and $V$ to not interact substantially in generating $\mathbf{X}$.

These assumptions are, of course, not universal. For example, in many prediction settings, the label $Y$ is more plausibly caused by the input $\mathbf{X}$. This situation is often referred to as a causal prediction setting. One example of a causal prediction task occurs when predicting the risk of cardiovascular disease ($Y$) from environmental risk factors ($\mathbf{X}$) in settings where a sensitive attribute affects those environmental risk factors. In this case, $Y$ could be a function of blood pressure ($\mathbf{X}^*$), which is affected by environmental risk factors, which are in turn affected by the the sensitive attribute. In this setting, the relationship between fairness and robustness criteria differs substantially from the relationships we outline below (see discussion in Section 4.2). Similarly, in many anti-causal prediction settings, there is substantial interaction between the sensitive group $V$ and the target label $Y$ in generating the input $\mathbf{X}$. For example, if $Y$ represents a heart condition, $V$ is the patient's age, and $\mathbf{X}$ is a waveform from an electrocadiogram (ECG), the presentation of heart conditions in an ECG are very different depending on whether the patient is a child or an adult. In this case, our characterization of the optimal risk invariant predictor (introduced below) would be different.

Finally, when we discuss the properties of learning algorithms, we make an overlap assumption on the source distribution, $P_s$. Specifically we assume that the support of $P_s(V)P_s(Y)$ is contained in the support of $P_s(V, Y)$. Intuitively, this assumption implies that we observe all combinations of $Y$ and $V$ during training time that will also appear in any target test distribution. Absent such an assumption, the behavior of $f$ on unobserved combinations is unlearnable using the observed data.

## 3.1 Robustness as Risk Invariance under Shifts in Dependence

Here, we review some robustness criteria that have been introduced before in our anti-causal setting, with the goal of drawing a formal connection to fairness criteria. Robustness criteria and fairness criteria often have distinct motivations. While fairness is often motivated by a principle of non-discrimination, robust or invariant predictive methods are often motivated by generalization to out-of-distribution settings.

Of particular interest here are scenarios where a "shortcut" association between inputs and target label changes between training and deployment time (Geirhos et al., 2020). A canonical example of shortcut learning is presented in the context of classifying animals from images in Beery et al. (2018), where classifiers often failed when animals in the test set appeared in new and unusual locations: for example, cows, which frequently appeared in grassy locations in the training set, were misclassified when they appeared on beach backgrounds in the test set. While any model might be prone to shortcut learning, recent empirical findings have shown that deep neural networks are especially prone to relying on shortcuts (Beery et al., 2018; Sagawa et al., 2019; 2020; Ilyas et al., 2019; Azulay & Weiss, 2018; Geirhos et al., 2018; D'Amour et al., 2020). A flurry of recent papers suggest methods to create robust models that give predictions that are invariant to the spurious correlations or shortcuts (Sagawa et al., 2019; Creager et al., 2021; Arjovsky et al., 2019; Krueger et al., 2021).

In this paper, we focus on a particular notion of robustness called risk invariance. Following the robustness literature, we assume that the model $f$ is trained on a source distribution $P_s$, and measure its predictive risk on one or many target distributions $P_t$. We write the generalization risk of a function $f$ on a distribution $P$ as $R_P = \mathbb{E}_{\mathbf{X}, Y \sim P}[\ell(f(\mathbf{X}), Y)]$, where $\ell$ is a predictive loss (we define this as the logistic loss in arguments that follow). We say that a model $f$ is risk invariant across a family of distributions $\mathcal{P}$ if its predictive risk is the same for each target distribution $P_t$ in that family. Here, we consider families of target distributions that can be generated from $P_s$ by interventions on the causal DAG in Figure 1. Specifically, we consider interventions on the dependence between $Y$ and $V$ that keep the marginal distribution of $Y$ constant. [1] Each distribution in this family can be obtained by replacing the source conditional distribution $P_s(V \mid Y)$ with a target conditional distribution $P_t(V \mid Y)$:

$$\mathcal{P} = \{P_s(\mathbf{X} \mid \mathbf{X}^*, V) P_s(\mathbf{X}^* \mid Y) P_s(Y) P_t(V \mid Y)\}, \tag{1}$$

This family allows the marginal dependence between $Y$ and $V$ to change arbitrarily.

For our analysis, one distribution contained in the set $\mathcal{P}$ will be important: the distribution where $Y \perp\!\!\!\perp V$, i.e., $P^\circ := P_s(\mathbf{X} \mid \mathbf{X}^*, V) P_s(\mathbf{X}^* \mid Y) P_s(Y) P^\circ(V)$. We refer to $P^\circ$ as the idealized distribution. In the chest x-ray example, $P^\circ$ is the distribution where a pneumonia patient is equally likely to be female or male, and similarly a healthy patient is equally likely to be female or male. We formally define this notion of robustness next.

**Definition 1** (Risk Invariance)**.** *Given the family $\mathcal{P}$ defined in equation (1), we say that a set of predictors $\mathcal{F}_{Rob}$ is risk invariant if they have the same risk for all $P_t \in \mathcal{P}$, i.e., if $\mathcal{F}_{Rob} = \{f : R_{P_t}(f) = R_{P_t'}(f) \quad \forall P_t, P_t' \in \mathcal{P}\}$.*

Risk invariance arises naturally as a criterion for checking whether a model's prediction depends on "shortcut" features in $X$ that are influenced by $V$ but not by $Y$. In our setting, any predictor that does not rely on such shortcuts will necessarily be risk invariant (Makar et al., 2021; Veitch et al., 2021).

## 4 Fairness and robustness in the anti-causal setting

In this section, we discuss connections between fairness and robustness that are specific to our anti-causal setting. These connections inform considerations about how practitioners may wish to choose between using an unconstrained model, or enforcing separation or independence. In particular, we show that there is substantial alignment between risk invariance and separation in this setting, which can provide motivation for preferring separation over an unconstrained model or a model that enforces independence.

### 4.1 Separation and Predictive Performance

Often the most difficult choice in deciding to apply fairness criteria to a classification problem is whether to constrain the model at all. Some of the oldest discussions in applying fairness to machine learning [2] center on tradeoffs between parities enforced by fairness-constrained models and overall predictive performance of unconstrained models (see, e.g., Corbett-Davies et al., 2017; Mitchell et al., 2021). Here, we revisit this discussion in our anti-causal setting, and show that separation can be motivated on purely performance-oriented grounds if the notion of performance is expanded to include predictive risk under distribution shifts. In addition, we show that the optimal risk invariant predictor in this setting satisfies separation, suggesting that algorithms that target the optimal risk invariant predictor can be effective for learning models that are fair according to the separation criterion.

Our discussion here revolves around two results. First, in the anti-causal setting in Figure 1, separation implies risk invariance. Secondly, the optimal risk invariant predictor in this setting satisfies separation. We present these two results formally in turn, then discuss their implications. We include the proofs of these

---

[1] This notion of risk invariance could be generalized to include cases where the marginal distribution of $Y$ also changes, but would introduce some notational overhead. It would require that a re-weighted risk be invariant across such a family.

[2] See also, discussions that pre-date machine learning, such as discussions of fairness in education assessment as reviewed in Hutchinson & Mitchell (2019).

statements to highlight how the results depend on our core causal assumptions (anti-causal structure and the sufficiency of $\mathbf{X}^*$).

**Proposition 1.** *In the anti-causal setting shown in Figure 1, suppose that a a predictor $f$ satisfies separation in the training distribution, that is, $f(\mathbf{X}) \perp\!\!\!\perp_{P_s} V \mid Y$. Then $f$ is risk-invariant with respect to the family of target distributions $\mathcal{P}$ defined in* (1).

*Proof.* We show this by decomposing the risk of the model $f$ on any target distribution $P_t \in \mathcal{P}$ in terms of the conditional risk given $V$ and $Y$. Within the family $\mathcal{P}$, $P_t(\ell(f(\mathbf{X}), Y) \mid Y = y, V = v)$ is the same for all $P_t \in \mathcal{P}$; thus, we can write the risk conditional on $Y$ and $V$ independently of the target distribution. Let $R_{vy} := \mathbb{E}_{P_t}[\ell(f(\mathbf{X}), Y) \mid V = v, Y = y]$ for any $P_t \in \mathcal{P}$. The overall risk on a target distribution $P_t$ can be written as the weighted average of these subgroup risks

$$R_{P_t} = \sum_{y \in \{0,1\}} P_s(Y = y) \left[ P_t(V = 0 \mid Y = y) R_{0y} + P_t(V = 1 \mid Y = y) R_{1y} \right]. \tag{2}$$

Now, the separation criterion states that $f(\mathbf{X}) \perp\!\!\!\perp V \mid Y$, which implies $\ell(f(\mathbf{X}), Y) \perp\!\!\!\perp V \mid Y$, which implies that $R_{0y} = R_{1y}$ for $y \in \{0, 1\}$. Thus, the terms in the summation (2) are the same regardless of the $P_t(V = v \mid Y = y)$ factors, and thus the risk is invariant for all $P_t \in \mathcal{P}$. $\qquad \square$

**Proposition 2.** *In the anti-causal setting shown in Figure 1, (a) the optimal risk invariant predictor with respect to $\mathcal{P}$ has the form $f^*(\mathbf{X}) = \mathbb{E}[Y \mid \mathbf{X}^*]$,* [3] *and (b), this predictor satisfies separation.*

*Proof.* Part (a) is shown as Proposition 1 in Makar et al. (2021). The key points of the proof are that (1) $\mathbb{E}[Y \mid \mathbf{X}^*]$ is representable by a function $f^*(\mathbf{X})$ under the assumptions made in Section 3 (namely, the assumption that $\mathbf{X}^*$ can be written as $e(\mathbf{X})$); (2) under the uncorrelated distribution $P^\circ$, $\mathbb{E}[Y \mid \mathbf{X}^*]$ is Bayes optimal; and (3) $\mathbb{E}[Y \mid \mathbf{X}^*]$ is risk invariant. (2) and (3) imply that no other risk invariant predictor can have lower risk across $\mathcal{P}$.

Part (b) follows from the fact that $\mathbf{X}^*$ is d-separated from $V$ conditional on $Y$ in the DAG in Figure 1. Thus, $\mathbb{E}[Y \mid \mathbf{X}^*] \perp\!\!\!\perp_{P_t} V \mid Y$ for all $P_t \in \mathcal{P}$. $\qquad \square$

*Remark* 1. Veitch et al. (2021) also provide a number of related results in their study of the implications of counterfactual invariance in anti-causal settings. Counterfactual invariance requires that the predictions of a model be invariant across label-preserving counterfactual versions of the input, such as the counterfactual that we would observe if the sensitive attribute were different. Our findings concern a narrower case where there exists a sufficient statistic $\mathbf{X}^*$, which Veitch et al. (2021) refer to this as the "purely spurious" case. In their findings, Veitch et al. (2021) show that counterfactual invariance implies the separation criterion generally in anticausal settings, and that the optimal counterfactually invariant predictor in also minimax optimal in the purely spurious case. Here, our results speak more directly to the implications of fairness criteria in the purely spurious setting, and, by focusing on the weaker risk invariance property, we make the connection without the conceptual ambiguity of defining counterfactuals with respect to sensitive attributes (see, e.g. Kohler-Hausmann, 2018, for discussion of this point).

These results have several implications in the motivation and practical application of the separation criterion. First, they provide a counterpoint to standard discussions of fairness-performance tradeoffs. In particular, they highlight the importance of specifying the distribution (or distributions) on which fairness and performance are defined. On one hand, in our setting, the Bayes optimal in-distribution predictor, $\mathbb{E}_{P_s}[Y \mid \mathbf{X}]$, does not in general satisfy separation (this follows because $\mathbf{X}$ is not d-separated from $V$, so $\mathbb{E}[Y \mid \mathbf{X}]$ is not independent of $V$). This implies that a model satisfying separation must have lower-than-optimal in-distribution accuracy. On the other hand, because Proposition 1 shows that separation implies risk invariance, the performance of a model that satisfies separation cannot degrade if the model is deployed in a causally consistent scenario included in $\mathcal{P}$. And in fact, as Proposition 2 shows, the best possible risk invariant model satisfies separation, implying that there is no tradeoff between optimal *invariant* performance and this fairness criterion. Thus, if model performance beyond the training distribution $P_s$ is important in an application,

---

[3]Note that $\mathbb{E}_{P_t}[Y \mid \mathbf{X}^*]$ is the same for all $P_t \in \mathcal{P}$, so we omit the distributional subscript on this expectation.

risk invariance can provide a purely performance-oriented motivation for enforcing a separation criterion. We note in passing that since separation and sufficiency are at odds, risk invariant models in the anticausal settings will not satisfy sufficiency.

Secondly, Proposition 2 suggests that, in our anti-causal setting, criteria designed for targeting optimal risk invariant predictors may be effective for learning classifiers that satisfy separation, even if they do not imply separation directly. For example, consider the approach proposed in Makar et al. (2021) that targets the optimal risk invariant predictor by enforcing an *ideal distribution independence* (IDI) criterion.

**Definition 2** (Ideal Distribution Independence). *A model $f$ satisfies IDI iff $f(\mathbf{X}) \perp\!\!\!\perp_{P \circ} V$; i.e., if its predictions $f(\mathbf{X})$ are independent of the sensitive attribute $V$ under the ideal distribution $P^\circ$, where $Y \perp\!\!\!\perp_{P \circ} V$.*

While the IDI criterion does not itself imply separation, Makar et al. (2021) show that enforcing IDI at training time can lead to more efficient recovery of the optimal risk invariant predictor, which does satisfy separation. This suggests that enforcing risk invariance in practice can also help to close fairness gaps. We demonstrate this empirically in our chest X-ray application presented in section 6.

### 4.2 Separation versus Independence

It is useful to note that considering the robustness properties of a predictor in its particular causal setting also distinguishes between different fairness criteria. In particular, in non-trivial instances of our anti-causal setting, there is a conflict between the optimal risk invariant predictor and independence.

**Proposition 3.** *Let $f^*(\mathbf{X}) = \mathbb{E}[Y \mid e(\mathbf{X})] = \mathbb{E}[Y \mid \mathbf{X}^*]$ be the optimal risk invariant predictor with respect to $\mathcal{P}$. In addition, assume that $\mathbb{E}[f(\mathbf{X}^*) \mid Y = y]$ is a non-trivial function of $y$, i.e., that the value of $Y$ actually affects the expectation of the sufficient statistic $\mathbf{X}^*$. Then for any distribution $P_t \neq P^\circ$ in $\mathcal{P}$, $f^*(\mathbf{X}) \not\perp\!\!\!\perp V$ and independence is not satisfied.*

*Proof.* Note that for each $v$, $\mathbb{E}_{P_t}[f^*(\mathbf{X}) \mid V = v] = \sum_{y \in \{0,1\}} \mathbb{E}[f^*(\mathbf{X}) \mid Y = y] P_t(Y = y \mid V = v)$. For any $P_t$ where $Y \not\perp\!\!\!\perp V$, the weights $P_t(Y = y \mid V = v)$ in this summation will differ as a function of $v$. By assumption, the expectations $\mathbb{E}[f^*(\mathbf{X}) \mid Y = y]$ differ for different values of $y$, so changing their weights in the summation will change the sum. Thus, for $v \neq v'$, $\mathbb{E}_{P_t}[f^*(\mathbf{X}) \mid V = v] \neq \mathbb{E}_{P_t}[f^*(\mathbf{X}) \mid V = v']$, which implies $f^*(\mathbf{X}) \not\perp\!\!\!\perp V$. $\square$

If downstream performance of a model on shifted distributions is an important consideration in an application, this result provides motivation against using independence as a fairness criterion, because it implies a tradeoff with the best achievable invariant risk. We demonstrate the practical implications of this tradeoff in our chest X-ray example in section 6.

Importantly, we note that this conflict between the risk of a model on downstream applications and independence can change depending on the causal structure of the problem. In particular, Veitch et al. (2021) show that in some causal prediction settings, where $V$ causes $\mathbf{X}$, but $\mathbf{X}$ causes $Y$, independence is compatible with the optimal predictor that satisfies a related robustness criterion that they call counterfactual invariance (Veitch et al., 2021, see Theorem 4.2). This highlights the importance of considering causal structure when deriving the implications of enforcing particular fairness criteria in a given application. However, we leave further discussion of the causal prediction scenario, and the related implications for independence and counterfactual invariance criteria for future work.

## 5 Enforcing Separation

The discussion in section 4 implies that there are two ways to enforce separation during training. First, it can be enforced directly by encouraging equality between representation distributions conditional on $Y$. Alternatively, as shown in section 4, it can be enforced indirectly by minimizing predictive risk subject to the IDI criterion, by encouraging equality between the marginal distributions of learned representations $\phi$ under the ideal distribution $P^\circ$. In this section, we discuss the practical considerations around different implementation schemes and highlight that learning algorithms that enforce separation directly face distinct

technical challenges from those that encourage separation through enforcing the IDI robustness criterion. Because of this distinction, as we show in Section 6, the IDI-motivated algorithm can achieve better separation in practice.

We center this discussion on a specific family of approaches that relies on estimates of distributional discrepancies to enforce statistical independences. Specifically, we focus the Maximum Mean Discrepancy (MMD) since it is a popular choice in the fairness literature (Prost & Qian, 2019; Madras et al., 2018; Louizos et al., 2015), and the robustness literature (Makar et al., 2021; Guo et al., 2021). The MMD defined as follows:

**Definition 3.** *Let $Z \sim P_Z$, and $Z' \sim P_{Z'}$, be two arbitrary variables. And let $\Omega$ be a class of functions $\omega : \mathcal{Z} \to \mathbb{R}$, $\mathrm{MMD}(\Omega, P_Z, P_{Z'}) = \sup_{\omega \in \Omega} \left( \mathbb{E}_{P_Z} \omega(Z) - \mathbb{E}_{P_{Z'}} \omega(Z') \right)$.*

When $\Omega$ is set to be a general reproducing kernel Hilbert space (RKHS), the MMD defines a metric on probability distributions, and is equal to zero if and only if $P_Z = P_{Z'}$. We take $\Omega$ to be the RKHS and drop it from our notation. MMD-based regularization methods enforce statistical independences at training time by penalizing discrepancies between distributions that would be equal if the independence held. The MMD penalty can be applied to the final layer $f$ or to the learned representation $\phi$. Both methods induce the required invariances. We follow the literature in imposing the penalty on the representation $\phi$.

The strategy for directly enforcing separation penalizes discrepancies in representation distributions conditional $Y$ (Prost et al., 2019). Such a strategy is translated to a learning objective as follows:

$$h_{\text{C-MMD}}, \phi_{\text{C-MMD}} = \operatorname{argmin}_{h,\phi} \frac{1}{n} \sum_{i=1}^{n} \ell(h(\phi(\mathbf{x}_i)), y_i) + \alpha \cdot \sum_y \widehat{\mathrm{MMD}}^2 (P_{\phi_{0,y}}, P_{\phi_{1,y}}), \tag{3}$$

where $P_{\phi_{v,y}} = P(\phi(\mathbf{X})|V = v, Y = y)$, $\alpha$ is a parameter picked through cross-validation, $\widehat{\mathrm{MMD}}^2$ is an estimate of $\mathrm{MMD}^2$. In the experiments below, we use the V-statistic estimator for the MMD presented in Gretton et al. (2012). This estimator relies on the radial basis function (RBF), which requires a bandwidth parameter $\gamma$ picked through cross-validation.

While this strategy is straightforward, it has some practical limitations, especially when training using mini-batches of data in stochastic gradient descent. Within each batch, C-MMD requires first dividing the population based on $Y$ then estimating the MMD within each subgroup. This effectively reduces the sample size used for MMD estimation, making the estimates more volatile and less reliable, especially when batch sizes are small.

Meanwhile, a strategy for enforcing separation indirectly through IDI, developed by Makar et al. (2021), makes use of a weighted marginal MMD discrepancy (WM-MMD). WM-MMD requires a marginal estimate of the MMD computed with respect to $P^\circ$ rather than the observed $P_s$. This strategy uses reweighting to manipulate the observed data such that it mimics desirable independencies in $P^\circ$. Specifically, it uses the following weights: $u(y, v) = \frac{P_s(Y=y)P_s(V=v)}{P_s(Y=y,V=v)}$, such that for each example, $u_i := u(y_i, v_i)$. This weighting scheme maps expectations under $P_s$ to expectations under $P^\circ$. The final objective becomes:

$$h_{\text{WM-MMD}}, \phi_{\text{WM-MMD}} = \operatorname{argmin}_{h,\phi} \sum_{i=1}^{n} u_i \ell(h(\phi(\mathbf{x}_i)), y_i) + \alpha \cdot \widehat{\mathrm{MMD}}^2 (P_{\phi_0^{\mathtt{u}}}, P_{\phi_1^{\mathtt{u}}}), \tag{4}$$

where $\widehat{\mathrm{MMD}}^2 (P_{\phi_0^{\mathtt{u}}}, P_{\phi_1^{\mathtt{u}}})$ is a weighted estimate of the MMD. This strategy also has practical challenges. While WM-MMD does not require this data-slicing, if base rates are too skewed within groups, the weights may be highly variable, and introduce volatility into the regularization. This is a typical risk with weighted estimators (see, for example (Cortes et al., 2010)).

Ultimately, the better strategy to employ to enforce separation depends on the context. For example, we find that WM-MMD is far more stable in our experiments on chest X-rays, and that the instability due to data-slicing induced by C-MMD leads it to exhibit a fairness-performance tradeoff in practice that WM-MMD is less prone to. This highlights the value of expanding the toolbox of fairness regularizers to include regularizers targeted at causally compatible robustness criteria. For more general use, we present a concrete heuristic to choose between the two methods in a given application.

By comparison to those two approaches, an approach that encourages independence rather than separation can be implemented by penalizing the unweighted prediction loss and the unweighted marginal MMD as follows:

$$h_{\text{M-MMD}}, \phi_{\text{M-MMD}} = \operatorname{argmin}_{h,\phi} \frac{1}{n} \sum\nolimits_{i=1}^{n} \ell(h(\phi(\mathbf{x}_i)), y_i) + \alpha \cdot \widehat{\text{MMD}}^2(P_{\phi_0}, P_{\phi_1}). \tag{5}$$

## 6 Experiments

In this section, we empirically demonstrate the implications of the relationships between fairness criteria and robustness criteria in an anti-causal setting. In this demonstration, we show that several of the arguments made at the population level in Section 4 bear out in a practical application. Specifically, following Jabbour et al. (2020), we consider the task of predicting pneumonia ($Y$) from chest x-rays ($\mathbf{X}$) considering sex to be a protected attribute ($V$). In this context, we demonstrate the close practical connection between separation (as measured with EO) and risk invariance. First, models that better satisfy EO exhibit greater risk invariance (Proposition 1). Secondly, we show that training procedures that target risk invariance can yield models that satisfy EO (Proposition 2), and somewhat surprisingly, they perform *better* than models that explicitly trained to satisfy separation. Third, no such alignment holds between independence (as measured with DP) and robustness (Proposition 3).

**Data and Test Distribution Design.** We conduct this analysis on a publicly available dataset, CheXpert (Irvin et al., 2019). The data contain 224,316 chest x-rays of 65,240 patients. Each chest x-ray is associated with 12 labels corresponding to different pulmonary conditions. Each label encodes if the corresponding condition is confidently present, confidently absent or might be present with some uncertainty. We select patients who have pneumonia (confidently present), or have no finding (i.e., all conditions are confidently absent). For the former group, we give them the label pneumonia (i.e., $y = 1$), for the latter group we give them the label "healthy" (i.e., $y = 0$). We exclude patients who do not have pneumonia but might have other pulmonary conditions from our analysis since other conditions such as Atelectasis or Lung Edema can be visually similar to pneumonia, making them hard to distinguish from pneumonia without additional information (e.g., medical charts). This leaves us with a total of 21,232 unique x-ray studies. We split the dataset into 70% examples used for training and validation, while the rest is held out for testing.

At training time, we undersample patients recorded as women who did not have pneumonia to create setting where a naïve predictor might have systematic errors for the under-sampled group. Specifically, we sample the data such that 30% of the population has pneumonia, i.e., $P_s(Y = 1) = 0.3$, and the majority of women patients have pneumonia whereas the majority of men patients do not have pneumonia i.e., $P_s(V = 1|Y = 1) = P_s(V = 0|Y = 0) = 0.9$. In this setting, as we show later in figure 2, a deep neural network trained in the usual way learns to rely on sex of the patient to predict pneumonia.

Here, the family of target distributions is the family of distributions that are compatible with the DAG in Figure 1, where the base rates of pneumonia were systematically skewed between sex groups by selective sampling. This set includes all distributions where $P_t(V = 1 \mid Y = y)$ is allowed to take on any value between 0 and 1 for each $y$. To test our models, we generated 6 test distributions from this family. We denote these test distributions $\mathcal{P}_t = \{P_{0.1}, P_{0.3}, P_{0.5}, P_{0.7}, P_{0.9}, P_{0.95}\}$, where $P_\mu$ is generated such that $P_t(V = 1 \mid Y = 1) = P_t(V = 0 \mid Y = 0) = \mu$ with $P_t(Y = 1)$ held constant.

**Implementation.** We use DensNet-121 (Huang et al., 2017), pretrained on ImageNet, and fine tuned for our specific task. We use DenseNet because it was shown to outperform other commonly used architectures on the CheXpert dataset (Irvin et al., 2019; Ke et al., 2021; Jabbour et al., 2020). All models are implemented in TensorFlow (Abadi et al., 2015). We train all models using a batch size of 64, and image sizes $256 \times 256$ for 50 epochs. We used publicly available code provided by the authors in Makar et al. (2021)[4].

In addition to C-MMD, WM-MMD described in section 5, we implement a deep neural network (DNN) without any robustness or fairness penalties. We note that C-MMD is the same as MinDiff, which is the approach to enforce EO in Tensorflow[5].

---

[4]https://github.com/mymakar/causally_motivated_shortcut_removal
[5]https://www.tensorflow.org/responsible_ai/model_remediation

For the three MMD based models, we need to pick the free parameter $\alpha$ which controls how strictly we enforce the MMD penalty, and $\gamma$, which is the kernel bandwidth needed to compute the MMD. We follow the cross-validation procedure outlined in Makar et al. (2021). Specifically, we split the training and validation data into 75% for training and 25% for validation. We further split the validation data into 5 "folds". We compute the MMD on each of the folds. The MMD is computed the same way as in training, meaning for the conditional MMD penalty, we compute the conditional MMD on each of the 5 folds, and similarly for the weighted marginal MMD, and the marginal MMD. We then compute the mean and standard deviation of the MMD estimate over the 5 folds. Using a T-test, we test if the mean MMD is statistically significantly different from zero. We exclude hyperparameters that yield a non-zero MMD. Out of the remaining hyperparameters, we pick the model that has the best performance, i.e., the lowest logistic loss. For the DNN, we perform L2-regularization. We pick the regularization parameter based on the validation loss. Additional details about hyperparameters are included in the appendix.

## 6.1 Results

**Sex as a shortcut in pneumonia prediction**
In our first set of results, we examine the extent to which each model utilizes information about sex to detect pneumonia. To do this, we measure the performance of each model across our set of shifted test sets $\mathcal{P}_t$. Models that encode information about sex to predict pneumonia (i.e., use sex as a shortcut) will experience a degradation in performance on test distributions where the marginal correlation between label $Y$ and sex $V$ has changed. Meanwhile, a model that does not use this shortcut should have invariant performance across these test sets.

For each of the distributions in $\mathcal{P}_t$, we compute the the area under the receiver operating curve (AUROC) of each predictor on each of the test distributions as a measure of its performance across distribution shifts. To estimate the uncertainty in the AUROC, we create 1000 bootstraps of the test set, and compute the means and standard deviations of the AUROC.

Figure 2 shows the results of this analysis. The $x-$axis shows $P(V = 1 \mid Y = 1) = P(V = 0 \mid Y = 0)$ at test time, while the $y-$axis shows the corresponding mean AUROC. The vertical dashed line shows the conditional probability at training time. Notably, DNN, which does not incorporate any fair-

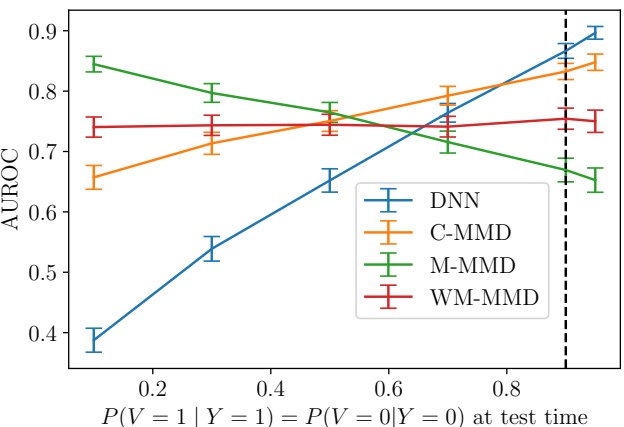

Figure 2: Test distribution on the x-axis, and performance measured by AUROC on the y-axis. Dashed line shows training distribution. DNN, which does not incorporate any fairness/ risk invariance criteria has the best in-distribution performance but its performance quickly deteriorates across distribution shifts. M-MMD over-penalizes the correlation between $Y, V$. While WM-MMD achieves worse in-distribution performance compared to DNN and C-MMD, it has has better out of distribution performance for distributions most dissimilar to the training distribution.

ness/ risk invariance criteria has the most severe dependence on the shortcut, and by relying on sex as a shortcut, it achieves the best in-distribution performance. However, such performance quickly deteriorates on test distributions where the strength of the shortcut association is smaller or even "flipped". While C-MMD should not, in theory, encode information about sex, we see that its practical implementation yields some dependence on sex, albeit less severe than that of the DNN. As we show later in this section, this somewhat surprising behaviour of C-MMD is because of its poor finite sample properties, which arise due to data-slicing. Meanwhile, M-MMD *uses* the shortcut to satisfy the fairness criterion, which leads to encoding an opposite dependence on sex. Finally, while WM-MMD achieves worse in-distribution performance compared to DNN and C-MMD, it has has better out of distribution performance for distributions most dissimilar to the training distribution. This suggests that WM-MMD has the least dependence on sex as a shortcut compared to the other predictors.

**Alignment between fairness and robustness in anti-causal settings.** Here, we examine the extent to which different fairness criteria (separation/independence) and their implementation align with robustness across distributions.

We empirically measure robustness by computing:

$$\text{Robustness} = \big| \max_{P_\mu \in \mathcal{P}_t} R_{P_\mu}(f) - \min_{P_\mu \in \mathcal{P}_t} R_{P_\mu}(f) \big|,$$

where $R_{P_\mu}$ is the logistic loss achieved on the distribution $P_\mu \in \mathcal{P}_t$. Lower values imply higher robustness. We measured the violations to the EO and DP criteria by generating a test set that is drawn from the same distribution as the training set, and computing:

$$\text{EO violation} = \max_y \big| \mathbb{E}_{P_s}[f(\mathbf{X})|Y = y, V = 1] - \mathbb{E}_{P_s}[f(\mathbf{X})|Y = y, V = 0] \big|, \quad \text{and}$$

$$\text{DP violation} = \big| \mathbb{E}_{P_s}[f(\mathbf{X})|V = 1] - \mathbb{E}_{P_s}[f(\mathbf{X})|V = 0] \big|.$$

Figure 3 (left, middle) show our measure of robustness on the x-axis. Each subplot shows a different fairness criterion on the y-axis. To estimate the uncertainty in the fairness and robustness metrics, we create 1000 bootstraps of the test set, and compute the means and standard deviations of each of the metrics. Both subplots confirm that the DNN which does not incorporate any fairness/robustness penalties indeed encodes information about sex in the learned representation, and is neither robust nor fair by any metric of fairness. In addition, the results confirm our analysis in section 4: we see that better robustness (i.e., risk invariance) tracks with less violations to the EO criterion in figure 3 (left). The same is not true for the DP criterion in figure 3 (middle): M-MMD achieves lower violations to the DP criterion compared to WM-MMD yet the former is less robust than the latter. While both C-MMD and WM-MMD are expected to perform favorably, we see that WM-MMD consistently performs better on robustness and fairness. We revisit this comparison in detail below.

**Risk invariance and its empirical relationship to EO and DP.** Here, we seek to empirically validate the analysis presented in section 4. Recall from Definition 1 that our robustness criterion, risk invariance, requires that a model achieve the same risk (i.e., the same logistic loss) on test sets sampled according to different distributions with varying correlation between pneumonia and sex. To measure risk invariance empirically, we computed the logistic loss on each of the test distributions in the set $\mathcal{P}_t$ (defined above), and measured risk invariance by computing:

$$\text{Robustness} = \big| \max_{P_\mu \in \mathcal{P}_t} \hat{R}_{P_\mu}(f) - \min_{P_\mu \in \mathcal{P}_t} \hat{R}_{P_\mu}(f) \big|,$$

where $\hat{R}_{P_\mu}$ is the logistic loss on the distribution $P_\mu$. Lower values imply higher robustness. We also measured the violations to the EO and DP criteria by generating a test set that is drawn from the same distribution as the training set, and computing:

$$\text{EO violation} = \max_y \Big| \mathbf{E}_{P_s}[f(\mathbf{X})|Y = y, V = 1] - \mathbf{E}_{P_s}[f(\mathbf{X})|Y = y, V = 0] \Big|, \quad \text{and}$$

$$\text{DP violation} = \Big| \mathbf{E}_{P_s}[f(\mathbf{X})|V = 1] - \mathbf{E}_{P_s}[f(\mathbf{X})|V = 0] \Big|.$$

Figure 3 shows our measure of robustness on the x-axis. Each subplot shows a different fairness criterion on the y-axis. The results are also presented in a table format in the appendix. We estimate the uncertainty in the fairness and robustness metrics by bootstrapping the test set, as described before. All subplots confirm that the DNN which does not incorporate any fairness/robustness penalties indeed encodes information about sex in the learned representation, and is neither robust nor fair by any metric of fairness. In addition, the results confirm our analysis in section 4: we see that better robustness (i.e., risk invariance) tracks with less violations to the EO criterion in figure 3 (left). The same is not true for the DP criterion in figure 3 (right): M-MMD achieves lower violations to the DP criterion compared to WM-MMD yet the former is less

robust than the latter. We note that despite the fact that the robustness of M-MMD and WM-MMD seems overlapping, a simple two sided ttest rejects the null (that the robustness of the two models is the same) with p-value $< 0.001$

While both C-MMD and WM-MMD are expected to perform favorably, we see that WM-MMD consistently achieves better on robustness and fairness suggesting that the latter is more stable. We will revisit this comparison in detail later.

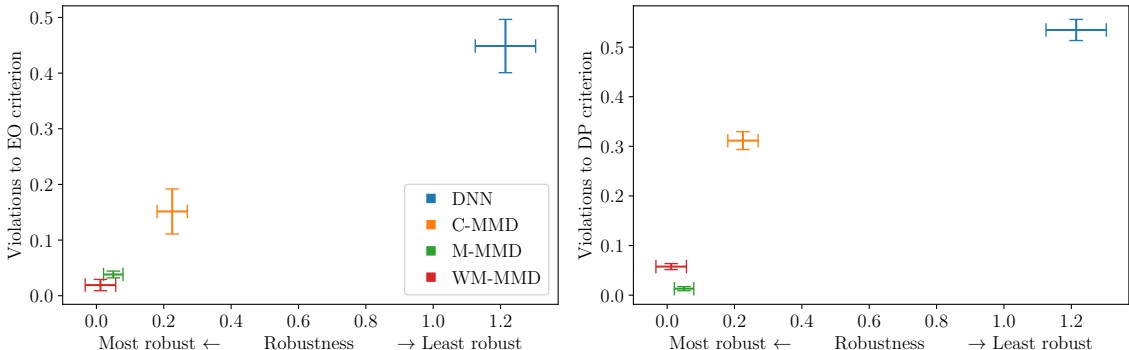

Figure 3: Robustness on the x-axis, and different fairness criteria on the y-axis. Left plot shows violations to EO. Right plot shows violations to DP. Models with higher robustness achieve lower fairness violations for EO, but not DP which is consistent with the analysis in section 4. The simple DNN relies on sex as a shortcut achieving the worst robustness and fairness properties. WM-MMD outperforms C-MMD achieving better robustness and lower EO violations.

**WM-MMD is more stable than C-MMD.** We now investigate the surprising discrepancy in performance between WM-MMD and C-MMD: while in theory they both target risk invariance, empirical performance suggests that WM-MMD is more effective.

This discrepancy can be explained, in part, by how well each optimized optimized MMD criterion generalizes to test data. Figure 4 shows the estimated MMD on the training, validation and testing data broken down by the target label. Here the testing data is sampled from the same distribution as the training data to highlight estimation error rather than errors due to data shift. The plot shows several important findings. First, the estimated MMD at training time is a more reliable estimate of the test-set MMD when weighted marginal MMD penalties are used at training and validation time, signaling that data slicing in C-MMD leads to unreliable estimates. Second, the difference between the MMD

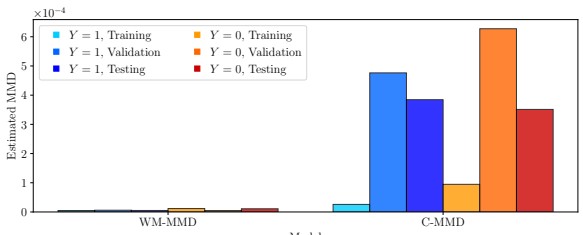

Figure 4: Estimated conditional MMD on the training, validation and testing data for WM-MMD and C-MMD. Estimates of the MMD are more stable for the WM-MMD objective than C-MMD.

among the group defined by $Y = 1$ compared to the group defined by $Y = 0$ is smaller when using WM-MMD. Smaller difference in the MMD between the two groups is important to ensure that the outcomes for both groups defined by the target label do not vary based on the protected attribute.

This analysis can be repeated in practice to choose between the two penalties: the estimated MMD on the validation data is a reliable proxy for the estimate's generalizability. In practice, if there is a large discrepancy between the training and validation C-MMD estimates, WM-MMD might be a better option and vice versa.

## 7 Discussion

In this work, we showed that by taking into account the specific causal structure of a prediction problem, we are able to draw deep connections between notions of robustness and fairness. We established a practical near-equivalence between separation and risk-invariance in an anti-causal prediction setting. Specifically, in one direction, separation implies risk invariance, while in the other direction, algorithms that are used to obtain performant risk invariant predictors yield predictors that approximately satisfy separation. This connection provides performance-oriented motivation for applying the separation criterion in an important class of problems, and provides a new set of tools, borrowed from the robustness literature, to enforce the criterion in practice.

More generally, our findings demonstrate that the understanding causal structure of a given problem can yield useful practical insights, both for understanding how different modeling constraints impact the real world behavior of models, and for implementing more sample efficient models that comply with practitioners' stated goals. As models are deployed in more complex settings, causality can be a useful tool for motivating, expressing, and satisfying ever-more detailed practitioner requirements.

### Broader Impact Statement

This work is aimed to help practitioners and researchers reason about one aspect of the broader impact of ML models, particularly those that are constrained to satisfy certain statistical parities. This work could provides motivation for enforcing parities like separation when there is reason to believe that the causal assumptions hold. However, this motivation is necessarily incomplete, and should serve as one of many different factors when considering how to manage the broader impact of an ML system. In particular, fairness and other socially salient goals should be determined based on the context of the application, including how the model influences real decision processing, the potential for harm, and how tradeoffs interact with existing social disparities.

### Acknowledgments

We are thank the anonymous reviewers, and Victor Veitch for their feedback. This work was partially funded by NSF grant No. 2153083. Any opinions, findings, and conclusions or recommendations expressed in this material are those of the author(s) and do not necessarily reflect the views of the National Science Foundation.

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
