# OpenReview forum: "Fairness and robustness in anti-causal prediction"
_TMLR — Accepted by TMLR_

### Review · Reviewer_9GZG · 2022-10-09

**Summary Of Contributions:**

This paper investigates the connections between “separation” fairness criterion and “risk invariance” robustness notion under anti-causal settings. The authors theoretically give the relationships of robustness and fairness in Proposition 1 and Proposition 2. They also empirically validate their findings on the task of detecting pneumonia from X-rays and show that robustness-motivated approaches can be used to enforce separation, and that they often work better in practice than methods designed to directly enforce separation.

**Audience:**

Yes

**Claims And Evidence:**

Yes

**Requested Changes:**

Please address my previous concerns

**Strengths And Weaknesses:**

Strengths:
1. The work provably gives the connections between the non-discrimination criterion, separation, and the risk invariance.
2. It’s good to consider practical situation of detecting pneumonia from chest X-rays across different population subgroups defined by a sensitive attribute to validate the results.
3. They also discuss the relationship between risk invariant and independence.

Weaknesses:
1. I have some concern about the contribution from method side since the authors use the MMD method from [1].
2. The non-discrimination fairness criteria typically include independence, separation and sufficiency in [2]. The authors consider the relationship between robustness and independence. But how about the relationship between robustness and sufficiency.

---

> ### Author Response · Authors · 2022-10-28
> **Contribution and sufficiency**
>
> We thank the reviewer for their thorough review!
>
> **1- Contribution:** We stress that the novelty of our work lies in establishing a connection between robustness in the anticausal setting and fairness metrics rather than suggesting a new approach. In establishing this connection, our work expands the toolbox that is available for enforcing fairness criteria to include recent approaches suggested for the purpose of robustness.
>
> **2- Sufficiency:** We studied only separation and independence because they are more dominant in the fairness/ML literature. However, by virtue of the fact that sufficiency and separation cannot both hold in non degenerate settings, this means that robustness methods in the anticausal settings will not lead to models that satisfy sufficiency (similar to the independence setting). We will add that in writing.

---

### Review · Reviewer_QUzX · 2022-10-10

**Summary Of Contributions:**

This paper evaluates two popular fairness measures in the transportability setting, including the equalized odds  (Hardt et al., 2016), called separation in this paper, and the demographic disparity, called independence. The authors consider an anti-causal setting, which is graphically described in Fig. 1, where $Y$ is the true label, $X$ is the observed input features, V is the sensitive attribute, and $X^*$ is an latent sufficient statistics summarizing the causal influence of Y on X. The goal of the leaner is to obtain an optimal predictor $\hat{Y} \gets f(X)$ that could estimate the true label $Y$ given input feature $X$ while satisfying the pre-specified fairness constraints.

The novelty of the authors' treatment is that they consider the fairness prediction in the transportability setting, where the generative distribution associated with the sensitive attribute $P(V|Y)$ could be different in the target domain. The challenge is to learn a risk invariant predictor $f(X)$ whose predictive performance is robust across different parametrizations of $P(V|Y)$. The authors prove that it is generally infeasible to obtain an optimal risk invariant predictor while satisfying independence. However, such an optimal predictor is still obtainable under the separation constraint. Simulation results corroborate the authors' findings.

**Audience:**

Yes

**Broader Impact Concerns:**

The authors have included a discussion addressing the broader impact of this paper. Overall, this work helps researchers understand the challenges of fair ML in practical settings, particularly those that are constrained to satisfy certain statistical parities. This work could provides
motivation for enforcing parities like separation under the anti-causal setting.

**Claims And Evidence:**

Yes

**Requested Changes:**

Could the authors provide more details on the experiments? Particularly, could the authors provide a table containing detailed values of fairness violations and robustness measures in Figure 3?

**Strengths And Weaknesses:**

This paper is well-organized and clearly written. The authors study a novel setting of fair supervised learning in the transportability setting. The authors analyzed two popular fairness measures and reached a concise, yet interesting conclusion. That is, an optimal robust predictor is generally obtainable under separation but not independence. Given the popularity of these two common fairness measures and the relevance of transportability in many practical settings, this paper could be helpful for AI researchers across disciplines.

As for the weakness, more details could be included fo experiments. For instance, the authors claim that "we see that better robustness (i.e., risk invariance) tracks with less violations to the EO criterion in figure 3 (left). The same is not true for the DP criterion in figure 3 (right)." However, in Figure 3, the plots of M-MMD and WM-MMD are very close to each other, seemingly within the margin of error. It is unclear if this simulation is direct evidence corroborating the authors' conclusion.

---

> ### Author Response · Authors · 2022-10-28
> **Similar performance between M-MMD and WM-MMD**
>
> We thank the reviewer for their thoughtful feedback
>
> **Figure 3, similarity between M-MMD and WM-MMD results:** The apparent similarity between the two models in terms of violations to the fairness criteria is an artifact of the visualization. Because of the poor performance of the unconstrained DNN, the y-axis in the two plots is expanded, making the M-MMD and WM-MMD look closer to each other. The reviewer is correct, however, that the difference in robustness that the robustness estimates are overlapping. However, a simple two sided ttest rejects the null (that the robustness of the two models is the same) with p-value < 0.001.
>
> We agree with the reviewer, however, that it might be helpful to include the following table of the results from figure 3.
>
> | Model  | Robustness   | Violations to EO criterion | Violations to DP criterion |
> | ------ | ------------ | -------------------------- | -------------------------- |
> | C-MMD  | 0.23 (0.045) | 0.15 (0.041)               | 0.31 (0.018)               |
> | WM-MMD | 0.01 (0.045) | 0.02 (0.01)                | 0.06 (0.006)               |
> | M-MMD  | 0.05 (0.029) | 0.04 (0.006)               | 0.01 (0.004)               |
> | DNN    | 1.21 (0.09)  | 0.45 (0.048)               | 0.53 (0.021)               |

---

### Review · Reviewer_UgDX · 2022-10-17

**Summary Of Contributions:**

The paper connects robustness, in the sense of risk invariance over label-sensitive attribute correlations, and fairness, in terms of separation, for anti-causal prediction tasks. Specifically, the paper show two interesting results: a) a predictor satisfying separation is risk-invariant in this specific setting, b) the optimal risk invariant predictor in this setting satisfies separation. Additionally, the optimal risk invariant predictor does not satisfy independence for any label-sensitive attribute distribution except the uncorrelated one.
The paper uses WM-MMD proposed by (Makar et al., 2021) to show that an Ideal Distribution Independence inducing loss with a good estimator of MMD can help to achieve separation and the proposed risk-invariance simultaneously. Specifically, the claim is that an effiicient risk-invariance inducing algorithm, such as WM-MMD, can help us to achieve good fairness performance, in terms of separation. The claim is empirically validated with experiments on the CheXpert dataset.

**Audience:**

Yes

**Broader Impact Concerns:**

The paper studies the connection between a specific sense of robustness and fairness, namely separation conditions. It is an interesting question for the community as developing fair predictors has social consequences, and training them efficiently and accurately is a natural question connected to it. But presently, the paper uses a risk-invariance inducing training mechanism to ensure separation. Given the theoretical and empirical evidence, it seems like a promising direction but it is hard to conclude anything more about efficiency/accuracy.

**Claims And Evidence:**

No

**Requested Changes:**

1. Technical: Please look into the weakness and corresponding questions.
2. Formatting: Except that there are minor typos, such as "my" -> "by", "excmaple"-> "example"

**Strengths And Weaknesses:**

Strength:
1. The connection  between robustness and fairness criterion are interesting and important questions to study.
2. The specific results on separation inducing training leading to risk-invariance and optimal risk-invariant predictor leading to separation are useful observations.
3. The paper is well-motivated, well-organised, and easy to read.
4. The paper talks about applicability and limitation of the anti-causal setting under construction. It also elaborates clearly the restricted family of distributions that it considers to study risk-invariance. It also justifies the choice.
5. The experiments are well-designed to proof the central hypothesis, i.e. the (near-)optimal risk-invariant predictor inducing separation.

Weakness:
1. The results are never rigorously proven, not even in the appendix. It is imperative to do so as they seem like the core contribution of the paper.
2. The paper uses a training mechanism inducing risk invariance to ensure separation. But such a result is never proven. As the proofs are not rigorous, it is not clear why or why not it can be proved that a predictor trained using risk invariance will (or will not) ensure separation. This is not trivial as the predictor obtained through WM-MMD is not the optimal risk-invariant predictor but an approximation of it. Then, the other question is if we have only a near-optimal risk-invariant predictor, how much can we lose in separation?
3. The paper never describes M-MMD in detail. It should be described clearly.
4. As M-MMD is said to improve independence, it is supposed to go worse separation. What is it we want to find from experiments except this intuitive fact?
5. The paper claims that risk-invariance inducing mechanisms induce better fairness performance than separation inducing algorithms. This claim is never validated as there is no comparison with any state-of-the-art separation inducing algorithms. It should be important to validate this claim.

---

> ### Author Response · Authors · 2022-10-28
> **Rigorous proofs and benchmarks**
>
> We thank the reviewer for the detailed feedback.
>
> **Results are never rigorously proven:** We make three main statements in propositions 1-3, all three propositions are proved in the main text, following the statement of the propositions.
>
> **“The paper uses a training mechanism inducing risk invariance to ensure separation. But such a result is never proven…This is not trivial as the predictor obtained through WM-MMD is not the optimal risk-invariant predictor but an approximation of it.”:**
> In proposition 2, we prove that the optimal risk invariant model also satisfies separation. Our proof is short since it largely builds on existing findings from Makar et al as well as basic principles of d-separation. The reviewer is correct in that our statement is asymptotic, and hence does not factor in estimation/approximation errors. Finite sample analysis which takes into account estimation errors has been studied elsewhere (e.g., see [Makar et al](https://proceedings.mlr.press/v151/makar22a/makar22a.pdf) and [Donini et al](https://proceedings.neurips.cc/paper/2018/file/83cdcec08fbf90370fcf53bdd56604ff-Paper.pdf)). Further analysis about the finite sample properties of the estimator are interesting future directions. However, we believe that our experimental results point to empirical evidence that our theoretical results bear out in a practical application.
>
> **Describe M-MMD in detail:** We excluded the exact objective function of M-MMD since it is identical to WM-MMD (equation 4) but it sets $u_i = \frac{1}{n}$. We will add the objective for M-MMD in the final version for clarity.
>
> **Better separation implies worse independence so why do we need to prove that:** We stress that the core message of our work is not that the relationship between two fairness approaches (in fact we never prove that separation implies worse independence). Rather, our work studies the relationship between fairness (as measured through separation or independence) and robustness in the anticausal setting. Our main finding is that separation and robustness are aligned, whereas independence and robustness are not.
>
> **No comparisons to the state of the art algorithms:**
> We note that the C-MMD baseline is the most common baseline used in the current literature. For example, see [Vietch et al](https://proceedings.neurips.cc/paper/2021/file/8710ef761bbb29a6f9d12e4ef8e4379c-Paper.pdf). In addition, C-MMD, which is the same as MinDiff, is the main technique for enforcing equalized odds in the official [tensorflow remediation library](https://www.tensorflow.org/responsible_ai/model_remediation), which represents a commonly accepted industry standard that is widely used in the machine learning community. We will highlight that in the writing.
>
> We thank the reviewer for identifying our typos, we will fix that in the final version.

---

### Review · Reviewer_XoML · 2022-10-23

**Summary Of Contributions:**

The research paper aims to draw a parallel between distribution shifts and fairness. The author assumes the input to be the function of the target label and sensitive attributes. The dataset used for the experiment is the X-RAY images dataset. The goal is to predict the presence of pneumonia wherein, also considering sex as one of the prevalent factors to ensure fairness mitigation. The author also provides a fresh perspective on fairness and performance tradeoffs. The author also compares his approach with MMD-based implementation.

**Audience:**

Yes

**Claims And Evidence:**

Yes

**Requested Changes:**

I think if you could add more explanation on MMD-based implementation, that would give readers a baseline to compare it against the right approach.

**Strengths And Weaknesses:**

Strengths

1. Explained the casual prediction setting in a detailed manner
2. Risk Invariance is explained well

Weakness
1. Some assumptions, such as pneumonia and patient sex influence changes, are highlighted well, but I don't see enough historical evidence that such assumptions are valid.

---

> ### Author Response · Authors · 2022-10-28
> **Assumptions regarding the effect of pneumonia and patient sex**
>
> We thank the reviewer for the thoughtful feedback!
>
> **Some assumptions, such as pneumonia and patient sex influence changes… [not] enough historical evidence that such assumptions are valid:**
> We wish to clarify that our assumption here is that the presence of pneumonia and the patient’s sex alters the appearance of the chest X-ray. First, the fact that pneumonia alters the chest X-ray is an established fact in the medical literature; the main diagnostic test for detecting the presence of pneumonia in a clinical setting is examining the chest X-ray (e.g., see guidelines [here](https://www.mayoclinic.org/diseases-conditions/pneumonia/diagnosis-treatment/drc-20354210)). If pneumonia did not alter the appearance of the chest X-ray, such a diagnostic test would be invalid. Second, the fact that the patient’s sex alters the chest X-ray is a consequence of physiological difference between different sexes (e.g., the appearance of breast tissue, and varying chest widths are well recorded in clinical literature)
> Our assumptions are corroborated by other studies in the machine learning literature (e.g., [Jabbour et al](https://sjabbour.github.io/files/jabbour20.pdf))
>
> **MMD explanation**:  We excluded the exact objective function of M-MMD since it is identical to WM-MMD (equation 4) but it sets $u_i = \frac{1}{n}$. We will add the objective for M-MMD in the final version for clarity.

---

### Decision · Action_Editors · 2022-12-05

**Recommendation:** Accept as is

**Comment:**

This work is about fair supervised learning under the transportability setting.
Two popular fairness measures have been analyzed, concluding that an optimal robust predictor is generally obtainable under separation but not independence. Given the popularity of these fairness measures and the relevance of transportability in many practical settings, this paper could be helpful for AI researchers across disciplines. The paper already incorporated some changes asked by the reviewers in their reviews, thanks to the responsiveness of the authors.

**Audience:**

All reviewers expressed interest in the main claim, which explains the relationship between fairness and robustness in the anticausal setting.

**Claims And Evidence:**

After discussion and modification, all reviewers agreed on the correctness and sufficing evidence. A  reviewer would like to have more comparisons with algorithms from EO SotA besides C-MMD, but is not requesting a precise one. Moreover, the core claim is not about performance but rather links between two popular measures.